# Simultaneous and stoichiometric purification of hundreds of oligonucleotides

Alessandro Pinto[1], Sherry X. Chen[1] & David Yu Zhang [1]

Purification of oligonucleotides has traditionally relied on mobility-based separation methods. However, these are imperfect, biased, and difficult to scale high multiplex. Here, we present a method for simultaneous purification of many oligonucleotides that also normalizes concentrations. The method uses a rationally designed randomer capture probe to enrich for oligos with perfect 5′ sequences, based on the observation that synthesis errors are correlated: product molecules with one or more deletions in one region are also more likely to have deletions in other regions. Next-generation sequencing analysis of 64-plex 70 nt purification products show a median 78% purity, a significant improvement over polyacrylamide gel electrophoresis and high pressure liquid chromatography (60% median purity). Additionally, 89% of the oligo products are within a factor of 2 of the median concentration.

[1] Department of Bioengineering, Rice University, Houston, 77005 TX, USA. Correspondence and requests for materials should be addressed to D.Y.Z. (email: dyz1@rice.edu)

High reagent purity is essential for understanding chemical reaction pathways and achieving high experimental reproducibility. Chemically synthesized DNA oligonucleotides (oligos) are an important class of reagents for genomics, biophysics, bionanotechnology, and biotechnology. Oligos serve as primers and probes for the amplification[1,2], enrichment[3], detection[4], and sequencing[5,6] of biological nucleic acids, and also serve as building blocks for self-assembled nanoscale materials with nanometer-precision[7,8]. In biotechnology, oligos are used as active drugs[9,10], templates[11], guides[12,13], or genomic building blocks[14].

Modern research advances use an ever-increasing numbers of oligo species: in genomics, a whole exome capture panel may consist of over 300,000 oligos each roughly 100 nt in length, and even small disease-specific targeted panels often contain hundreds of primer or probe oligos[15–17]. In DNA nanotechnology, DNA origami initially used roughly 300 distinct oligo staples[7], but have recently scaled to over 10,000 oligos[18]. Companies such as Roche Nimblegen, Twist Biosciences, and Agilent have developed chip-based oligo synthesis methods to produce pools of 5000–1,000,000 distinct oligo species, in order to meet these research needs. However, oligo purification technologies have not caught up with the scale and throughput of modern oligo demands.

Like any other chemical process, oligo synthesis is imperfect: during each nucleotide addition step, there is a small probability of either terminating the synthesis to result in a truncated oligo, or skipping over the nucleotide to result in an oligo that bears an internal deletion[19]. The error probability accumulates with each base, so the fraction of perfectly synthesized molecules of a desired sequence is significantly lower for longer oligos. For many applications where high purity oligos are desired, high pressure liquid chromatography (HPLC) or polyacrylamide gel electrophoresis (PAGE) post-synthesis purification is used to remove the majority of synthesis failures. However, the costs associated with such mobility-based purification methods dominate the overall oligo synthesis cost, rendering these approaches unsuitable for high-throughput applications. Application of HPLC or PAGE to a pool of oligos does not achieved the desired purification effect, because in the vast majority of use cases the oligos in the pool will have different mobilities due to length and sequence variations. Furthermore, even for single oligo purification, HPLC and PAGE struggle to completely remove oligo impurities that differ by only a single base from the intended sequence. Alternative purification technologies based on polymerization[20], mismatch recognition enzymes[21–23], or phase tags[24] have been proposed and demonstrated for simultaneous purification of pools of oligos, but these generally do not show improved purity over HPLC/PAGE, and the product sequences have not been systematically analyzed at the single-molecule level to characterize true purity.

Here, we introduce Stoichiometrically Normalizing Oligonucleotide Purification (SNOP), a method for simultaneously purifying hundreds of different oligos. In addition to improving oligo purity, the SNOP method also normalizes the concentrations of the oligo products, so that all products are at similar concentrations in the final pool. This feature of SNOP allows the user to overcome synthesis yield variations, and prevents a few oligo species from dominating a pool. In this manuscript, we experimentally demonstrate 64-plex and 256-plex SNOP, and characterized the product purities using next-generation sequencing (NGS). The high purity of SNOP products, combined with the high-throughput nature of SNOP, renders SNOP an attractive and affordable method for modern research and development needs.

## Results

**SNOP mechanism.** For each oligo product $O_i$, we design a corresponding precursor oligo $P_i$ with a tag sequence to the 5′ of the sequence of $O_i$ (Fig. 1a). The tag comprises three parts, from 5′ to 3′: a universal sequence that is common to all precursors, a precursor-specific barcode sequence, and a deoxyuracil (dU) nucleotide. Each barcode sequence consists of a number of alternating strong (G or C) and weak (A or T) nucleotides (e.g., CTCTCT or CAGACT). Barcodes sequences are designed this way to minimize the variation in standard free energy of hybridization to their complements. For the purposes of this manuscript, the length of this region $B$ is 6 or 8 nucleotides, corresponding to a total of $2^6 = 64$ or $2^8 = 256$ barcode instances.

During SNOP, the precursors are first hybridized to a biotinylated capture probe. The probe is synthesized as a randomer with degenerate nucleotides "SWSWSW" using a split-pool approach[25]; through the course of a single synthesis, all instances of the of the random sequences are created in roughly equal concentrations. Each precursor thus binds most favorably to one instance of the probe that is perfectly complementary to the precursor's tag. When the capture probe is the limiting reagent, the amount of each full-length precursor hybridized is similar despite variations in initial precursor concentrations. Subsequent solid-phase separation using streptavidin-coated magnetic beads removes unbound precursors, and applying USER enzyme mix (New England Biolabs) cleaves the oligo products from the tags at the dU site.

SNOP, in essence, selects for precursors with perfectly synthesized tag sequences in a multiplexed fashion. The entire SNOP protocol takes roughly 3 h, with 30 min being hands-on time and the remainder being waiting time for hybridization and digestion. This time is constant regardless of the number of different precursor and oligo product species, rendering SNOP significantly more scalable than traditional HPLC or PAGE post-synthesis purification methods.

The reason SNOP oligo products have higher purity than the precursors or directly synthesized oligo products is because DNA synthesis errors are correlated, rather than independent (Fig. 1b). Modern oligo sequence occurs in the solid phase on glass surfaces; imperfect uniformity of surface physics and chemistry result in some zones having higher nucleotide addition efficiencies than others. Consequently, an oligo molecule with a perfect 5′ tag sequence is more likely than average to have a perfect 3′ product sequence.

**Fluorescence studies.** We first study the purity of SNOP products bearing 3′ FAM-label, using fluorescence PAGE to visualize the length distribution of any impurities. For clarity, we first performed SNOP on a single precursor species $P_1$. The other 63 tags were also introduced in the input mixture each at equal concentration to the precursor. This was done to minimize non-selective binding of other regions of $P_1$ to the other capture probe instances (Fig. 2a), and to minimize nonspecific binding to the magnetic beads. The fluorescence PAGE results (Fig. 2b) show the length distribution of FAM-labeled oligos in the precursor $P_1$, in the intermediate $I_1$ before USER cleavage at the dU site but after removal of unbound precursors, and in the final product oligo $O_1$. Also shown is the control oligo $C_1$ with the same sequence as $O_1$, but purchased as a commercially HPLC-purified oligo (Integrated DNA Technologies). The SNOP product has a visibly smaller fraction of impurities as compared to the HPLC-purified oligo.

Next, we demonstrate multiplex SNOP on precursors of different lengths, using three precursors of different lengths, along with the 61 unused tags (Fig. 2d). The fluorescent PAGE results show that the SNOP product mixture contains insignificant impurities from any of the three precursors (Fig. 2e). Because the oligo products have different lengths, gel band intensity analysis

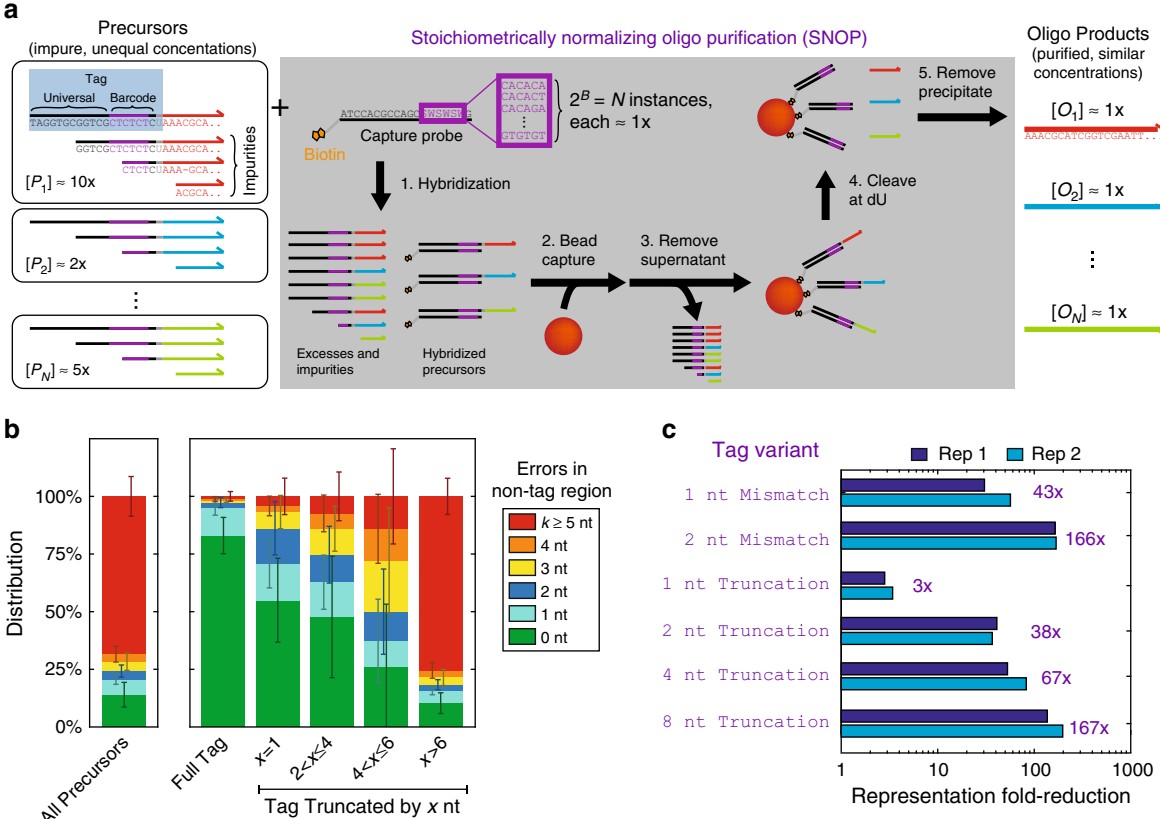

**Fig. 1** Stoichiometrically normalizing oligonucleotide purification (SNOP) concept and workflow. **a** The input reagents for SNOP are chemically synthesized oligonucleotide precursors $P_1$ through $P_N$ that contain imperfect synthesis products with 5′ truncations and/or internal deletions, and with potentially very different concentrations. SNOP produces a pool of oligonucleotide products $O_1$ through $O_N$ that has high fractions of oligos with perfect sequence, and with all products at roughly equal concentration. SNOP uses a single biotinylated capture probe oligonucleotide synthesized with a degenerate "SWSWSW" randomer subsequence. Each instance of the randomer is complementary to one precursor tag sequence. The different instances of the capture probe are all at roughly equal concentration, due to split-pool oligo synthesis. Precursors with perfect tag sequences hybridize to the probe and are captured by streptavidin-coated magnetic beads. Subsequent cleavage at the deoxyuracil (dU) site using the USER enzyme mix (https://www.neb.com/products/m5505-user-enzyme) releases the oligo products into solution. Setting the capture probe to be the limiting reagent allows all SNOP products to be all at roughly equal concentrations. **b** SNOP enriches the fraction of perfect oligos because synthesis errors are correlated; molecules with no truncations or deletions in the tag sequences are also more likely to not have any deletions in the oligo product sequence. Shown in this panel are NGS sequence analysis results of a pool of $N = 64$ precursor oligonucleotides; error bars show standard deviation across different oligos (see Methods for library preparation details). **c** SNOP is very sensitive to small sequence changes in the tag; even single-nucleotide variations result in significantly reduced binding yield (see also Supplementary Note). This property allows SNOP products to be both highly pure and stoichiometrically normalized

allows quantitation of relative concentrations. Starting from nominally equal concentrations of precursors, all three product concentrations are within 50% of each other.

Importantly, SNOP product concentrations remain relatively uniform (less than 3-fold variation) even when precursor concentrations differ grossly by 25-fold (Fig. 2e). The robustness of the relative product concentrations reflects the specificity of binding between precursors and their intended capture probe instances. To the extent that oligo product concentrations differ slightly based on grossly different precursor concentrations, some amount of nonselective binding between precursors and non-cognate capture probe instances likely still exist.

**Tag-oligo assignment for high-plex SNOP**. SNOP functions based on the specific hybridization of precursor barcodes to their complementary instances of the capture probe. Consequently, precursors should be designed so that their tags are accessible for hybridization, rather than part of intramolecular secondary structure. Mathematically, the reaction standard free energy of hybridizing each precursor to its cognate capture probe instance

$\left(\Delta G_{\mathrm{rxn}}^{\circ}\right)$ should all be similar. For example, a precursor with significant secondary structure involving its tag would have a significantly more positive $\Delta G_{\mathrm{rxn}}^{\circ}$, and would likely exhibit low hybridization yield, resulting in an under-representation of its oligo in the final SNOP product mixture (see Supplementary Note for extended discussion).

The design of SNOP precursor sequence comprises the design of the universal region and the assignment of barcodes to precursors. First, the universal region sequence is designed that has relatively low predicted binding to all desired oligo product sequences. Next, we consider the assignment of the different instances of the "SWSWSW" barcode to precursors. We start with a random bijection assignment, and then apply an iterative Monte Carlo algorithm to consider tag swaps, accepting the swaps that improve the overall $\Delta G_{\mathrm{rxn}}^{\circ}$ distribution of the entire panel (Fig. 3a). The optimization metric score S is calculated as the sum of the standard deviation of $\Delta G_{\mathrm{rxn}}^{\circ}$ and the range of $\Delta G_{\mathrm{rxn}}^{\circ}$.

We chose to use a randomized algorithm rather than systematically evaluating all $N^2$ potential assignments, in order to be more scalable to larger plex SNOP. Calculating $\Delta G_{\mathrm{rxn}}^{\circ}$ values have nontrivial runtimes for longer oligos. Figure 3b shows the

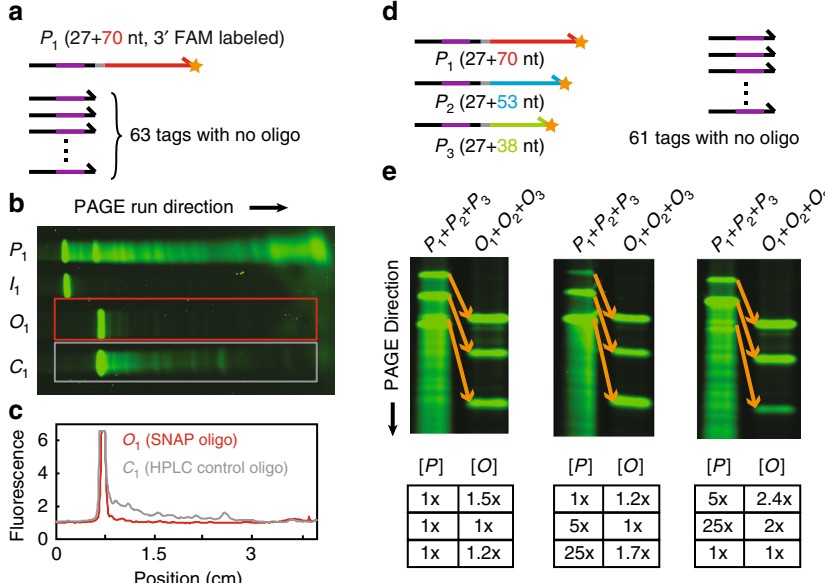

**Fig. 2** Fluorescence polyacrylamide gel electrophoresis (PAGE) analysis of SNOP products. **a** Precursors for analysis of SNOP product purity. One precursor that produces the oligo product of interest is synthesized with a 3′ FAM fluorophore label. Here, the randomer region has length $B = 6$, and the 63 other precursors contain only the tag sequence and does not produce an oligo product. This experiment thus shows the purity of a typical 64-plex SNOP product. **b** Fluorescent PAGE of the precursor $P_1$, the intermediate $I_1$ in which the tag is not cleaved, the oligo product $O_1$, and a control oligo $C_1$ ordered commercially with post-synthesis high pressure liquid chromotography (HPLC) purification. **c** Image analysis of the fluorescent gel indicates that the SNOP product $O_1$ has higher purity than the control oligo $C_1$. **d** Characterizing SNOP product stoichiometry using three FAM-labeled precursors of different lengths. **e** The relative concentrations of SNOP products are measured via the fluorescence intensity of each band. Despite gross differences in the precursor concentrations, SNOP product concentrations remain roughly 1:1:1

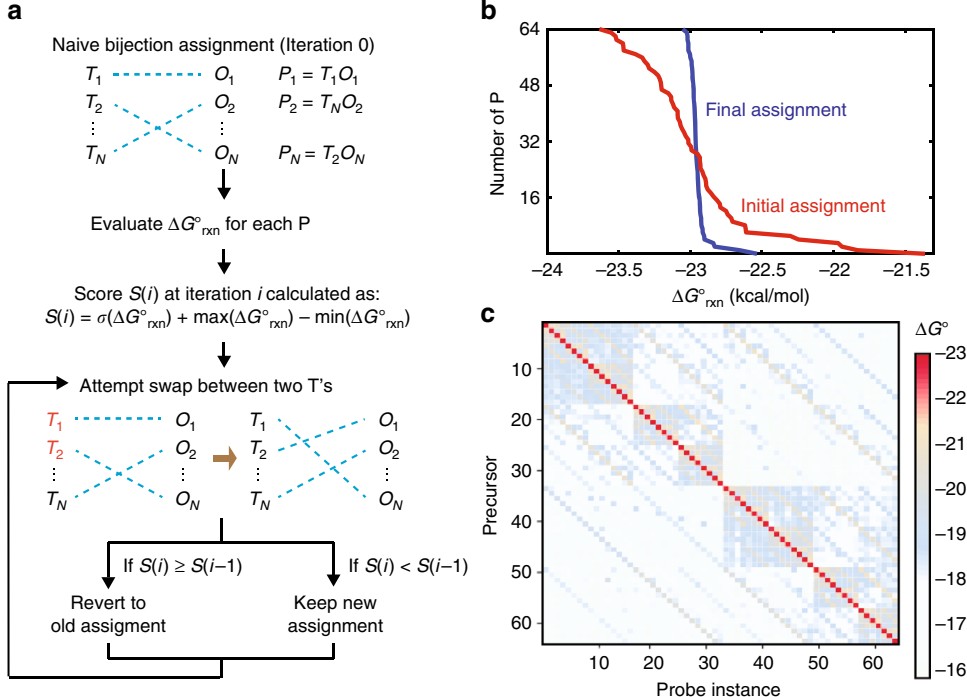

**Fig. 3** Precursor sequence design. Similar stoichiometry of SNOP products requires both (1) similar concentrations of probe randomer instances and (2) similar standard free energies of reaction $\left(\Delta G^\circ_{rxn}\right)$ between precursors and matched probe instances (see Supplementary Note for detailed explanation). **a** Tag-oligo matching algorithm to optimize uniformity of $\Delta G^\circ_{rxn}$. The time required for computing $\Delta G^\circ_{rxn}$ from sequence is nontrivial, so brute-force calculation of all possible pairings is not scalable. Instead, we perform Monte Carlo optimization using a fitness score $S$ calculated as the sum of the range and standard deviation of $\Delta G^\circ_{rxn}$. The algorithm is run for a fixed number of iterations (here 500), and the tag-oligo assignment set with the best $S$ is used experimentally. **b** Distributions of computed $\Delta G^\circ_{rxn}$ value for tag-oligo assignments before and after algorithmic optimization. **c** Predicted thermodynamics of hybridization between precursors and different probe instances for the optimized 64-plex tag-oligo assignments. The red diagonal corresponds to the hybridization of precursors and their matched probe instances. The lack of strong off-diagonal hybridization suggests that there will be little crosstalk interaction between precursors and unmatched probe instances

initial and final $\Delta G^{\circ}_{rxn}$ distributions for the 64-plex SNOP analyzed later in the manuscript. Through the course of 500 rounds of potential swaps, we arrived at a final barcode assignment that is more than 5-fold tighter than the initial assignment, with a $\Delta G^{\circ}_{rxn}$ range of about 0.6 kcal/mol. This uniformity helps ensure a minimal amount of nonselective binding between precursors and noncognate instances of the capture probe.

Figure 3c shows the distribution of calculated $\Delta G^{\circ}$ for intended and nonselective hybridization between tags and cognate or noncognate probe instances. From these values, the equilibrium binding of each precursor to each probe instance can be analytically determined. Although the multiplexed hybridization reaction likely does not reach equilibrium for all species, the fact that SNOP improves oligo purity strongly suggests that a significant degree of strand exchange is occurring. Put another way, the vast majority of probe molecules will initially encounter precursors that either are not matched to the probe instance, or harbor truncations/deletions in the tag region. We believe that a majority of these nonselective binding products are displaced by the cognate, full-length precursors during the course of the SNOP hybridization[26]. See Supplementary Note for in-depth discussion regarding the kinetic vs. thermodynamic regimes of SNOP.

**NGS analysis of SNOP product purity.** We next performed NGS experiments (Illumina MiSeq) to obtain more detailed information regarding the frequency and distribution of synthesis errors in both precursors and SNOP products, and to allow analysis of more highly multiplexed SNOP. To facilitate library preparation in a minimally biased manner, we designed all precursors to have a common 3′ sequence, which serves as a binding site for a common PCR primer (Fig. 4a). A single cycle of PCR is used with the common primer to make all SNOP products double-stranded, and then Illumina adaptors are appended using the NEBnext kit to create an NGS library.

NGS was run as an overlapped paired-end sequencing, and any paired reads in which the forward and reverse reads did not exactly match were bioinformatically discarded; this is done in order to minimize the effects of sequencing error on inferences regarding oligo purity. Remaining NGS reads were aligned to the product oligo sequences using Bowtie 2 end-to-end alignment, removing read artifacts from excess sequencing cycles. Subsequently, aligned reads in the SAM file were analyzed using custom Matlab code to determine the frequency and distribution of truncations, deletions, insertions, and replacements (for details see Supplementary Figures 4 to 12).

The purity of an oligo is here defined as the number of reads that perfectly match the intended sequence, divided by the

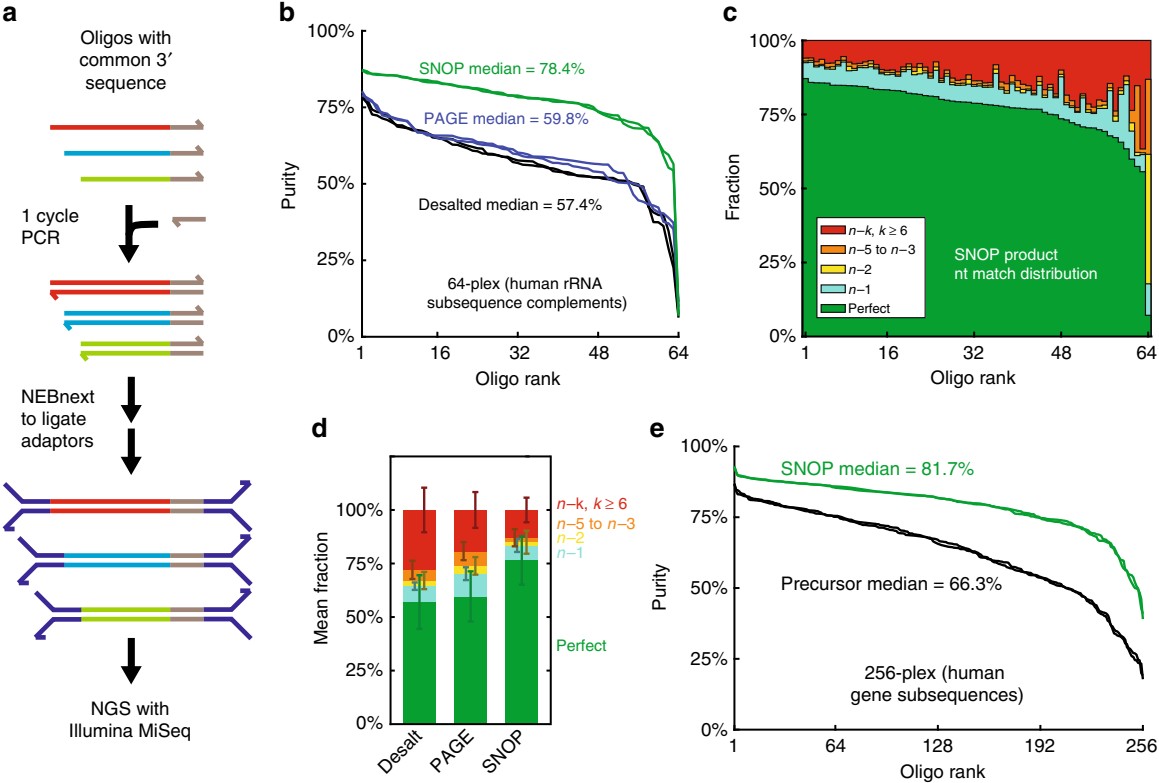

**Fig. 4** Characterization of high-plex SNOP product purities using next-generation sequencing (NGS). **a** To minimize the ligation and sequencing bias of library preparation and sequencing steps, the oligos used for this set of experiments all possess a common 3′ sequence. Using the complement of the common 3′ sequence as a PCR primer generates double-stranded product oligos. Afterwards, Illumina sequencing adaptors and P5/P7 sequences are appended using the NEBnext kit. In all NGS experiments, overlapped paired-end sequencing was performed, and any sequence in which the forward and reverse reads do not perfectly agree were discarded bioinformatically, in order to minimize the effects of NGS intrinsic error. **b** Distribution of purities observed for oligo products of a 64-plex SNOP; here, oligo products were all 70 nt long. Purity is here defined as the number of NGS reads that perfectly match the designed oligo sequence, divided by the number of NGS reads that align to the designed oligo sequence via Bowtie 2[29]. The purity distributions for two replicate libraries are plotted. SNOP products exhibit significantly higher purities than commercial oligos individually purified via PAGE; median purity values are displayed in the figure. **c** Summary of NGS reads aligned to each oligo, by the number of nucleotides matched to the designed sequence. For most oligos, the three most dominant classes of reads are Perfect (green), single-base truncations or deletions (cyan), or gross (6+) errors. **d** Summary of NGS reads analysis for the desalted, PAGE, and SNOP libraries. $N = 64$ precursor oligonucleotides; error bars show standard deviation across different oligos. **e** Distribution of purities observed for 256-plex SNOP products, as compared to commercial desalted oligos

number of reads that aligned to the intended sequence via Bowtie 2. Figure 4b summarizes the oligo purities of a 64-plex SNOP, in which each product is 70 nt long (94 nt precursors); the median purity out of the 64 SNOP products was 78.4%. For comparison, we also constructed a NGS library on independently synthesized desalted oligos, and another on independently synthesized PAGE-purified oligos; both of these show significantly lower purities, with medians of 57.4% and 59.8%, respectively. All NGS libraries were prepared and run in duplicate, and there is very-high reproducibility across the repeats.

Other than the universal 3′ sequence, the 64 oligo product sequences are complementary to subsequences of the human 5S, 5.8S, 18S, 28S, and mitochondrial ribosomal RNAs (see Supplementary Data File for oligo product sequences and origins). Thus, this 64-plex SNOP shows validation on a set of biologically relevant sequences, rather than "easy" sequences designed to be structure-free. The observed purities for these 64 oligos may be lower than typical for oligos of similar lengths, because human ribosomal RNA sequences have significant secondary structure that may reduce synthesis efficiency.

The purity values reported here based on NGS analyses are likely underestimates, because the library preparation process itself may introduce errors. For example, the primer used to bind the 3′ end of the products may itself have synthesis errors, and the 1-cycle PCR to make double-stranded amplicons may introduce errors[27]. Because the exact same protocol is used for all three libraries (SNOP, PAGE, and desalted), the quantitative amount of purity underestimation is likely the same. Consequently, we are confident that SNOP does produce significantly higher purity oligos than individual PAGE purification.

Analysis of the 64-plex SNOP product impurities (Fig. 4c) shows that, for most sequences, the dominant impurities are gross truncations/deletions and single-base deletions. There is one notable outlier with very low purity and a large fraction of 2 nt truncations at the 5′ end (sequence 35, see Supplementary Data File); this sequence is G/C rich and starts with "GGCGGGGGGG". The large amount of gross truncations remaining in the SNOP products is surprising, as such grossly truncated oligos should not remain after the SNOP process. Control experiments in which 5 alien sequences with no tag were added to the precursor mix indicated that there is only a minimal amount of nonspecific capture (Supplementary Figure 3). Consequently, we believe that these may be artifacts generated during the index addition step of the library preparation process.

When we bioinformatically exclude reads that harbor 6 or more nucleotide errors, the median purity of the SNOP products becomes 90.1% (Supplementary Figure 4).

One relatively surprising observation is that the individually PAGE-purified oligos exhibit only marginally higher purity than desalted oligos (Fig. 4b), and actually lower purity than the desalted oligos once we bioinformatically exclude the gross truncations and deletions (median 74.8% for PAGE, vs. 79.9% for desalted). These results suggest that the PAGE purification process actually damages the DNA oligos, increasing the fraction of oligos with 1 nt truncations or 1 nt internal errors. The overall purity of PAGE oligos is similar to that of desalted because the reduction of grossly truncated molecules is balanced by the increase of 1 nt errors. For many applications demanding high purity oligos such as gene synthesis or templating guide RNA production, such single-base errors can result in undesirable side products or off-target effects. In contrast, SNOP products exhibit both fewer gross truncations and fewer 1 nt deletions, as compared to desalted and PAGE oligos. Similarly, SNOP products are more pure than HPLC-purified oligos (Supplementary Figure 4).

We next observed purities for a 256-plex SNOP (using an 8 nt SWSWSWSW barcode); oligo products for this pool are complementary to subsequences of 183 expressed human mRNAs (Fig. 4e). The median purity for the 256-plex SNOP products is 81.7%.

If reads with gross truncations/errors are bioinformatically excluded, the SNOP product purity rises to 89.1%. Due to the high purification costs involved, we did not order these oligos as individually PAGE-purified oligos for purity comparison.

**NGS analysis of SNOP product concentrations**. We next analyzed our NGS data to determine the relative concentrations of the SNOP products. This process is nontrivial because there is known sequence-based bias both in library preparation (efficiency of ligating to adaptors) and in actual sequencing (flow cell capture efficiency and bridge PCR efficiency). We used the library of desalted oligos, with all oligos at nominally equal concentrations as indicated by the supplier, as a reference library. For other NGS libraries, the number of total reads for each oligo product is divided by the number of reads in the reference library to generate a nominal concentration. These nominal concentrations are then divided by the median of the nominal concentrations to generate relative concentrations. Mathematically,

$$[O_i] = \left( \frac{\text{Read}_i}{\text{Reads}_{i,D}} \right) \cdot C \qquad (1)$$

$$\text{Rel.Conc.}(O_i) = \frac{[O_i]}{\text{Median}_{i=1}^{N}[O_i]} \qquad (2)$$

where $[O_i]$ is the concentration of oligo product $O_i$, $\text{Reads}_i$ is the number of total aligned reads for $O_i$ in the library of interest, $\text{Reads}_{i,D}$ is the number of total aligned reads in the desalted reference library, and $C$ is a scaling constant common to all $O_i$ that does not require explicit definition.

To evaluate the suitability of this normalization method, we first analyzed the library of individually PAGE-purified oligos, which are also all at nominally the same concentration (Fig. 5a). The relative concentrations for the PAGE oligos are mostly with a factor of 2 of the median, but does exhibit a little variation, possibly due to either supplier quantitation error or biases in ligation/sequencing. We believe that the PAGE library relative concentration distribution represents the most uniform distribution that remains practically achievable.

The SNOP product library started with nominally equal concentrations of precursors, and shows slightly higher variation in relative concentrations than the PAGE library, with 89% of the oligos (57 out of 64) remaining within a factor of 2 of the median concentration. We attribute the expanded spread of relative concentrations to be primarily due to 3 factors, in order of importance: (1) unequal concentrations of different instances of the capture probe, (2) imperfections of $\Delta G^{\circ}_{\text{rxn}}$ prediction that result in some nonselective binding, and (3) differences in precursor purities that result in differing concentrations of capturable precursors. To rigorously validate the relative concentration distribution of the SNOP products, we also performed 64 individual digital droplet PCR[28] experiments using oligo-specific primers (Supplementary Figure 15 and Supplementary Discussion). Digital PCR results for the 64-plex SNOP products were largely consistent with the NGS results, indicating that 95% of SNOP products (61 out of 64) had concentrations within 2-fold of median. NGS results on the 256-plex SNOP product mixture shows a similar distribution of relative concentrations (Fig. 5b).

As designed, the SNOP process should produce roughly equal concentrations of products even when there are significant differences in precursor concentrations. To experimentally test

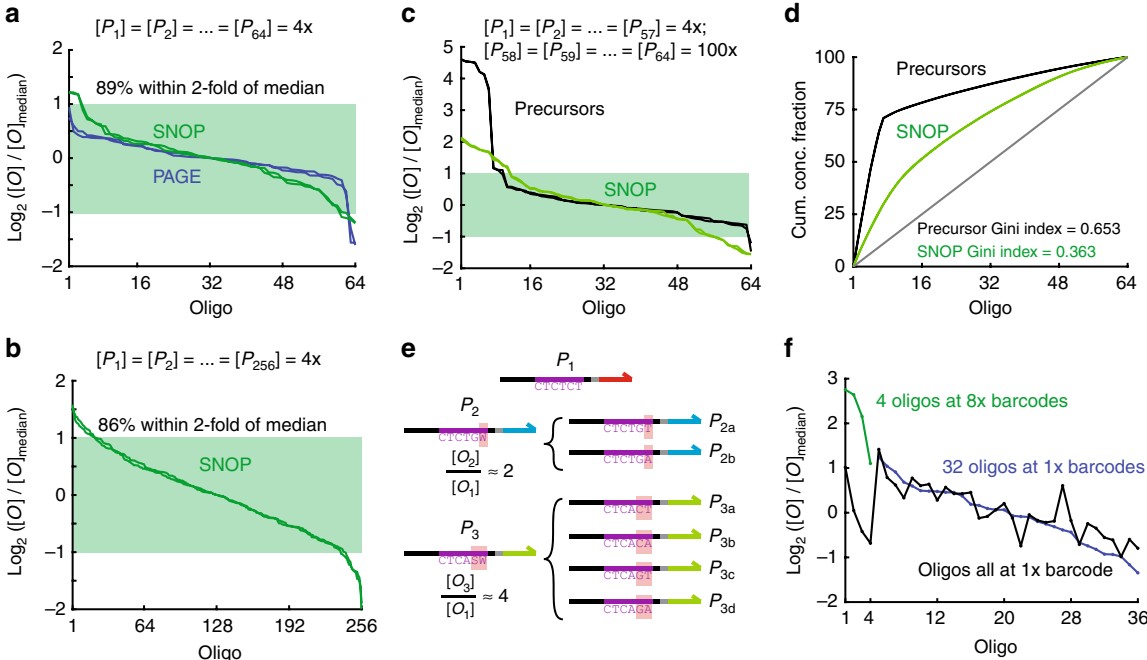

**Fig. 5** SNOP product stoichiometry. **a** Distribution of 64-plex SNOP product concentrations inferred from NGS results. Precursors were purchased commercially as desalted oligos and all precursors were mixed together at equal concentrations as the SNOP input. The desalted oligo library was used for NGS normalization to determine concentration from reads count. There is slight bias in the SNOP product concentrations, with roughly 90% of oligos within 2-fold of median concentration. **b** Relative concentration distribution of 256-plex SNOP. **c** Effects of gross biases in precursor concentrations on SNOP product stoichiometry. Seven of the 64 precursors were at 25-fold higher concentration compared to the rest, and these are reflected in the higher concentrations of the top 7 species in NGS for the precursor library. The SNOP product library shows a more equal distribution of oligo product concentrations, though the more skewed precursor concentrations does result in a wider range of product concentrations. **d** Lorenz curve showing the cumulative concentration distribution of the SNOP products from grossly biased precursors. From this figure, we determined that the SNOP process reduces the Gini inequality coefficient from 0.65 to 0.36. **e** Advanced stoichiometry control of SNOP products. Using degenerate nucleotides in the barcodes of some precursors allows the relative concentration of those precursors' products to be controllably increased. **f** NGS reads for 36-plex SNOP, in which 4 of the precursors had 8x barcodes (3 degenerate nucleotides). The black trace shows corresponding relative concentrations for the same species in the 64-plex SNOP library

this, we next started with a precursor mixture in which 7 of the 64 were at 25-fold higher concentration than the remainder (Fig. 5c). NGS of the precursors library reflects the roughly 25-fold higher concentration of the 7 species at the left. The relative concentrations of the SNOP products of these high species are just 2- to 4-fold above median, representing a significant (8-fold) reduction of bias. However, the concentration uniformity on the low end has also become worse, with 8 SNOP products now at between 2- and 4-fold lower than median, as compared to only 2 products when all precursor concentrations were roughly equal. This is likely because the higher concentrations of the 7 precursors increased the amount of nonselective binding between them and noncognate capture probe instances.

These results are generally consistent with our theory and analysis (see Supplementary Note): 25-fold precursor bias is likely the limit of what our current SNOP implementation can tolerate while still producing similar product concentrations. This limit is due to the thermodynamics of single-base mismatches, because two barcodes can differ by only a single nucleotide. SNOP robustness to precursor biases can be increased through more sparse usage of barcodes, e.g., if all pairs barcode sequences differed by at least 2 nucleotides.

The cumulative distribution plot of the relative concentrations (Fig. 5d) shows the overall degree of inequality, which can also be summarized with a single numeric metric, the Gini index. The value of the Gini index ranges between 0 (perfect uniformity for all species) and 1 (perfect inequality with only a single species represented in the SNOP product mixture). For this set of

precursors with seven gross excess species, the precursor Gini index was 0.653, and the SNOP product Gini index was 0.363. As a point of comparison, the nominally equal concentration PAGE oligos had a Gini index of 0.125, and the SNOP products from a nominally equal concentration mixture of precursors was 0.207 (see Supplementary Figure 15).

**Tunable increase of oligo concentrations via degenerate precursors.** Designing multiple precursors with the same oligo product is a simple way to increase the relative concentration of that species in the SNOP product mixture. This can be desirable for high-value targets where the user strongly wishes to avoid low relative concentrations. The length B of the barcode region controls the total number of different different precursors available ($B = 6$ for $2^6 = 64$-plex SNOP, $B = 8$ for $2^8 = 256$-plex SNOP). Subject to the above limitation on the total number of precursors, the number of precursors that can used to each oligo product species can be arbitrarily decided.

Individual synthesis of many different precursor molecules to generate the same product oligo is not cost-efficient. A more economical way to achieve higher relative concentrations of a set of desired oligos is to use degenerate randomer barcodes in the precursors (Fig. 5e). In this way, the number of precursor molecules synthesized scales only with the number of distinct product molecules. The tradeoff is that the number of barcodes assigned to each oligo product must a power of 2 (e.g., 1, 2, 4, 8, etc.). Given the distribution of relative concentrations for SNOP

products, we do not feel that finer control on number of barcodes assigned would be necessary.

Figure 5f shows NGS results for a 64-plex SNOP with 36 distinct oligo products; 32 of these have a single barcode, and four of these each have eight barcodes (3 nt randomer region). The four oligos with eight barcodes (green dots) show significantly higher relative concentrations than the other 32, though unexpectedly the concentrations are roughly 4-fold higher than median rather than 8-fold as designed. To facilitate comparison, the relative concentrations of a standard 64-plex SNOP with 64 product species are also plotted with the same $X$ indices. Oligo #4, which had a low (0.5x median) concentration in the standard SNOP, exhibits a high (2x median) concentration in the degenerate barcode SNOP. This demonstrates that low concentration oligo products can be "rescued" through the use of degenerate barcode precursors.

## Discussion

SNOP solves the emerging need in genomics, bionanotechnology, and biotechnology for many high purity oligos with reasonable labor and reagent costs. The randomer capture probe allows scalable and simultaneous purification many different oligos: although we demonstrated only up to 256-plex SNOP here, we imagine that scaling up further (e.g., to $2^{16} = 65536$-plex) will be technically straightforward. The SNOP product purities reported by NGS (median 78.4% for 64-plex, 81.7% for 256-plex) likely represent an underestimate of true purities, because the NGS library preparation process itself may introduce errors. Nonetheless, because NGS biases to purity analysis should be the same for all libraries, we are confident that SNOP products are significantly more pure than individually PAGE-purified oligos.

As far as we are aware, there have no been any previously demonstrated methods for multiplexed purification that normalizes concentration. This technology leverages the ability to split-pool synthesize DNA randomer probes with roughly equal representation of all instances. Based on our NGS analysis, our SNOP product concentrations were generally quite uniform, with 86–89% of oligo concentrations within 2-fold of median. Although this is not perfect, we believe that it is challenging to significantly further improve concentration uniformity, because even a pool of PAGE-purified oligos at nominally the same concentrations exhibit some concentration variation (95% within 2-fold of median). If desired, oligo species with low concentrations in the SNOP product mixture can be rescued via degenerate barcode precursors.

The design of SNOP precursors is somewhat technically complex, and the computational workflow is not yet optimized. A given set of desired oligo products may contain certain oligo species that cannot be co-purified via SNOP, e.g., if the oligos exhibit significant reverse complementarity to each other. Attempts to SNOP purify these would result in significantly lower purity, because the molecules captured on the magnetic bead may be due to "daisy-chain" binding via the product sequences. Large sets of desired oligos should be subdivided into subsets for purification, with all members of a subset being compatible for SNOP co-purification. Because we did not consider this when designing our 64- and 256-plex SNOP precursors, this and other nonspecific capture mechanisms likely contribute significantly to the impurities remaining the in the SNOP products in this work. There may be significant room for purity improvement through optimization of the experimental workflow (buffer, temperatures, and times) and of the computational design.

The purification yield was not a major concern or design criterion for this work; we typically used 4x excess of precursors relative to their cognate capture probe instances, so in the absence of nonselective binding our yield should be 25%. From our NGS and digital PCR results, we estimate that the yields for the 64-plex SNOP varied between 50 and 10%. Yield could be improved through decreasing the precursor excess, but doing so risks decreasing purity because excess capture probe instances will nonselectively capture precursors will imperfect tags.

## Methods

**Oligonucleotide reagents**. The sequences and synthesis details of oligonucleotides (oligos) used as precursors, reference species, capture probes, and primers are listed in Supplementary Data File. Oligos were synthesized by either Integrated DNA Technologies, or by Sigma-Aldrich Inc. Some reference species were individually post-synthesis purified via HPLC or PAGE, as annotated in the table. "Desalted" refer to oligos that were subject to standard desalting processes before lyophilization. Desalted oligos provided by Sigma were subject to a cartridge purification; desalted oligos provided by IDT did not include such a process. All oligos were quality-checked by the vendor via either capillary electrophoresis or mass spectrometry.

Oligos were delivered in "LabReady" format (100 μM in Tris-EDTA buffer). The 64-plex precursor oligos were pooled manually to form a stock solution roughly 1.6 μM in each species in Tris-EDTA buffer. The 256-plex precursor oligos were pooled manually to form a stock solution roughly 0.39 μM in each species in Tris-EDTA buffer. These stock solutions were stored at 4 °C.

**Hybridization buffer**. All hybridization reactions between the precursors and the capture probe occurred in a hybridization buffer that is 10 mM Tris-EDTA, 0.5 M NaCl, and 0.05% Tween-20 (volume/volume). Tris-EDTA and Tween-20 were purchased from Sigma-Aldrich, and NaCl stock solution (5 M) was purchased from Ambion.

**SNOP protocol**. In 64-plex SNOP, 6.4 μL of the precursor (10 pmol of each precursor) was mixed with 1.6 μL of the capture probe (160 pmol) and 92 μL of hybridization buffer. The mixture was kept for 2 h at 60 °C using an Eppendorf Mastercycler Personal instrument. Afterwards, 1 mg (100 μL) of Dynabead MyOne Streptavidin T1 magnetic beads (Thermo Fisher) were added. Before use, the magnetic beads were pre-washed three times in hybridization buffer. To ensure temperature uniformity, the beads were also pre-heated to 60 °C before addition to the precursor/probe mixture.

The resulting mixture was incubated for another 30 min at 60 °C to allow the biotin-streptavidin capture reaction to proceed. Next, the solution was placed in a Dynamag-96 side magnet (Thermo Fisher), and all magnetic beads and bound oligos were collected to one side of the microcentrifuge tube. Supernatant was manually removed using a pipettor, and the beads were washed six times: in each wash step, 200 μL of hybridization buffer was added, the solution was vortexed and then placed back in the Dynamag-96, and supernatant was removed.

2U (2 μL) of the USER enzyme mix (New England Biolabs) and 25 μL of the accompanying CutSmart buffer were added to the washed magnetic bead, and the solution was incubated at 37 °C for 1 h to allow cleavage at the dU site to release the oligo products from the magnetic beads. The supernatant solution of SNOP products was then extracted, and the magnetic beads were discarded.

**Fluorescent polyacrylamide gel electrophoresis**. For these experiments, we used 10% PAGE TBE-Urea precast Gels (Fisher Scientific). For each sample, roughly 2 pmol oligo was dissolved in 1x loading buffer, the latter comprising 48% formamide (v/v), 10 mM EDTA, and 1 mg/mL bromophenol blue. The mixture was then denatured by incubating at 95 °C for 10 min, and then loaded into the gel. Electrophoresis was run for 45 min at 60 °C at 110 V. The gel was then imaged using a Typhoon FLA 9500 gel scanner (GE Healthcare). Quantitation of gel band intensities was performed using accompanying software. Uncropped gel images are available in the Supplementary Information.

**NGS library preparation**. Supplementary Figure 1 shows an overview of the NGS library preparation process. In Step 1, 100 pmol of the common reverse primer was added to a total of 2.5 pmol of the SNOP products (or desalted/PAGE reference oligos), in 60 μL PCR buffer (Sigma Aldrich). This solution was heated to 95 °C for 5 min, and then held at 60 °C for 1 h to allow primer extension with Taq polymerase.

In Step 2, 10 units of polynucleotide kinase and 1 mM ATP were added, and the reaction mixture was incubated at 37 °C to allow 5′ phosphorylation. Next in Step 3, the NEBNext kit (New England Biolabs) was used to ligate on hairpin adaptors, following manufacturer protocols. In Step 4, index primers including Illumina P5/P7 adaptors were appended Finally in step 5, the library was individually SPRI purified using Agencourt AMPure XP beads (Beckman Coulter) following the manufacturer instructions. The purified library was checked for concentration using a Qubit dsDNA HS Assay Kit (Fisher Scientific) and for quality using a Bioanalyzer High Sensitivity DNA Analysis Kit (Agilent).

**NGS run and data analysis**. All NGS runs were performed on an Illumina MiSeq instrument, using 600 cycle v3 kits. In all NGS runs, overlapped paired-end sequencing was performed in order to minimize the effects of intrinsic sequencing error on interpretation of oligo purities.

NGS FastQ files were first pre-processed to remove adaptor sequences and artifact sequences past the end of the oligos using custom Python code. Subsequently, the trimmed reads were aligned to the oligo sequences using Bowtie 2 end-to-end alignment[29]. Consistently, 87 to 90% of the NGS reads in each library were aligned; the remaining reads likely correspond to adaptor or index dimers that were not fully removed by SPRI purification. The SAM files generated were then analyzed using custom Matlab code to compute statistics on oligo purity and relative concentration. Code is available upon request.

**Code availability**. NGS data summarized in the figures and custom Matlab 2016b code used to compute statistics on SAM files are available from the corresponding author upon request. Precursor sequence design software for the SNOP method can be made available to academic researchers under Non-disclosure Agreement.

**Data availability**. All data supporting the findings of this study are available from the corresponding author upon reasonable request.

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

## Acknowledgements

The authors thank J. S. Wang, J. Z. Fang, and J.X. Zhang for discussions and advice. This work was funded by NIH grant R01HG008752 to D.Y.Z.

## Author contributions

A.P. conceived the project, designed and conducted the experiments, analyzed the data, and wrote the paper. S.X.C. analyzed the data. D.Y.Z. conceived the project, guided experimental design, analyzed the data, and wrote the paper.

## Additional information

**Competing interests:** There is a patent pending for all authors on the SNOP method. D. Y.Z. is aco-founder and significant equity holder of Nuprobe Global. S.X.C. receives consulting fees from Nuprobe Global.

