## [Peer Review File · Nature Communications]

Reviewers' comments:

Reviewer #1 (Remarks to the Author):

Well written and methodologically complete. New method of oligo purification is presented and compares well to existing methods. Yet purpose of rather incremental increase of purity compared to other methods is not clear, especially with respect to pharmaceutical applications, where highest purity is required.

The effort in developing this method seems worthwhile, further improvement can be expected. Description of experimental protocols seems complete.

The acronym SNAP may interfere with identical acronyms which are eventually protected by trademarks.

Supplement "Methods"

page 1: SNAP protocol 2nd paragraph, last word should read "removed"

Reviewer #2 (Remarks to the Author):

Peer Review for Manuscript "Simultaneous and Stoichiometric Purification of Hundreds of Oligonucleotides"

By Jian-Sen Li, Associate Director, Scientific Research, Illumina, San Diego, CA, USA
12/26/2017

I am going to follow the general review guideline questions from Nature Communications for this manuscript.

•What are the major claims of the paper?

This paper describes a new approach to purify oligos of similar lengths in a claimed "high throughput" way (up to 256 oligos experiment wise), that can simultaneously yield higher purity (comparing to PAGE experiment wise) and a controlled oligo concentration range (within ~3 folds) despite of a large difference in raw oligos input (~25 folds). This approach entails the following steps:

1. Precursor hybridize to capture probe in mixture
2. Immobilize to Streptavidin coated magnetic beads
3. Wash
4. USER cut
5. Remove precipitate to yield final product

For the sake of discussion, let me just call this approach as hybe-immob-wash-cut approach.

•Are the claims novel? If not, please identify the major papers that compromise novelty

While this paper seems pretty novel, there is one publication that may compromise the novelty of this paper.

Highly parallel oligonucleotide purification and functionalization using reversible chemistry
Nucleic Acids Research, 2012, Vol. 40, No. 1, Kerri T. York, Ryan C. Smith, Rob Yang, Peter C. Melnyk, Melissa M. Wiley, Casey M. Turk, Mostafa Ronaghi, Kevin L. Gunderson, and Frank J. Steemers*, from Illumina

This paper described a "catch release" technology that can generate 5'-biotin functionalized oligos product with the following benefits

1. High throughput, up to 13000-15000 oligos, in comparison, the manuscript proved up to 256

oligos experimentally, further throughput increase may not be that easy (I will elaborate separately)

2. Scalable, up to umol for oligo pools, in comparison, the manuscript proved pmol for oligo pools.

3. Purification, by nature the catch release technology eliminates most of the oligo failures that do not have 5'-aldehyde mods, it may not eliminate internal deletion oligo failures that do have 5'-aldehyde mods, but such failures are largely dependent on raw oligo synthesis quality. In comparison, the manuscript claimed to purify off internal deletion oligo failures (to certain extent).

4. Equal concentration in the end product, in the paper and in our production, we have not evaluated experimentally if equal concentration in the end product is the case, but it does not matter to ILMN's exome enrichment application as the final 5'-biotin oligos are used in large excess to capture exome targets. In comparison, the manuscript claimed equal concentration in final product, I would like to understand more on why equal concentration is important with examples that matter to specific applications. In other words, why do we care about equal concentration?

5. Non-specific binding. Non-specific binding is not a concern to catch release paper due to high concentration UREA usage, in comparison, non specific binding might be a concern to the hybe-immob-wash-cut approach because (1) streptavidin coated magnetic beads are WELL known to have non-specific binding (2) undesired non-specific duplex formation could increase as throughput or plex increases

•Will the paper be of interest to others in the field? •Will the paper influence thinking in the field?

This paper can be interesting to others in the field and can influence thinking if the authors can specifically identify applications that are currently difficult to attain but would be otherwise easy to accomplish with the hybe-immob-wash-cut approach.

There are, however, several limitations in this manuscript that may limit its influence.

1. As the way it is right now, the 5' end cannot bear useful functionalization. If the authors switch to 5'-biotin capture probes (see below section), then 3'-end cannot bear useful functionalization, this could be a limitation. In comparison, the catch release technology can have both 3'-and 5'-functionalization separately, if required by certain applications.

2. This approach added 20 or more bases to the required synthesis. Thus, we expect it to be functionally limited to synthetic pools of <80 bp in length, possibly <60 bp in a practical sense.

3. Scale is limited with regard to synthesis pools. Likely this technique is useful only for smaller quantities, especially based on the necessity to oversynthesize b/c the expensive pull-down streptavidin coated magnetic beads are the limiting reagent.

4. The hybe-immob-wash-cut approach needs to use excessive oligos to saturate capture probes with duplex formation on desired targets, it is definitely not inexpensive, especially when scaling up, cost becomes a concern. In addition, the cost of USER enzymes can be a concern when scaling up.

5. Bioinformatic intensive for barcode design and no guarantee that cross-hybridizing will not be a factor for novel, new sequences.

•Are the claims convincing? If not, what further evidence is needed?

In general some claims seem reasonably convincing, for example, purity increase in general, although more control experiments would be needed.

The "scaling up is technically straightforward" seems questionable in discussion session. Another paragraph in discussion session may actually seem to explain why scaling up can be technically challenging.

•Are there other experiments that would strengthen the paper further? How much would they improve it, and how difficult are they likely to be?

I would like to get into technical details in this section:

1. 3'-biotin capture probe(s) preparation was specified as "made from IDT". I am going to assume it's made from 3'-biotin CPG in oligo synthesis. 3'-Biotin oligos made from 3'-biotin CPG are known

to have stability issues, the 3'-biotin may come off really quick due to hydrolysis in solution (I can explain in details if needed). That said, I wonder if the authors have experienced such instability issue. In general, efforts to reproduce the hybe-immob-wash-cut approach can be hard with a known stability issue. I suggest the authors try two things: (1) make 3'-biotin oligos from a different way (there are different ways) (2) try 5'-biotin capture oligos

2. The hybe-immob-wash-cut approach needs to use excessive oligos to saturate capture probes with duplex formation on desired targets, so N-1 impurity or N-2 impurity do not form duplex, but there are no experiments to prove that's the case. I would expect at least one experiment with N-1 spike in in the 1-plex experiment with LCMS analysis to show N-1 spike does not exist in final product.

3. For purity increase, it would be nice to add HPLC purified oligos (either reverse phase HPLC purified oligos and/or anion-exchange HPLC oligos) as one more control, in addition to PAGE purified oligos.

- Are the claims appropriately discussed in the context of previous literature?

Please see my previous comments about one publication missing from the manuscript.

- If the manuscript is unacceptable in its present form, does the study seem sufficiently promising that the authors should be encouraged to consider a resubmission in the future?

I am not sure if a resubmission in the future in the future should be considered for Nature Communications, mainly due to novelty compromised. If the authors could identify some particular applications that are currently different to attain but would be otherwise easy to accomplish with the hybe-immob-wash-cut approach. Such applications need to be fairly useful to most users in genomic field and/or biotech field.

Reviewer #3 (Remarks to the Author):

In the manuscript "Simultaneous and Stoichiometric Purification of Hundreds of Oligonucleotides," the authors introduce a novel method for the purification and balancing of pools of crude oligonucleotides. Purification beads are coated with a capture oligo containing a constant ("universal") dodecamer, a degenerate ("barcode") octamer, and a single RNA base (deoxyuracil). Crude oligo pools are synthesized that include the universal sequence at the 5' end. The pools are hybridized to the beads and then cleaved enzymatically. The carefully balanced degenerate octamer serves to balance the concentrations of the individual oligos in the pool. Testing of the method produced oligo purities around 80% with elements balanced in a two-fold range around the median. This is very good performance.

The discussion section compares the SNAP method only with PAGE-purification methods. It would be useful to also compare and contrast SNAP with enzymatic mismatch cleavage/removal methods. For example, the use of RecJ has been evaluated for use on conventional single-stranded oligos (Pubmed: 28459543), while MutS and other enzymes have been used for error-correction on double-stranded products built from conventional and microarray oligos (Pubmed examples: 15561997, 28733633).

The figures are all clear, appropriate, and support the conclusions of the paper. Perhaps the scale on supplemental figure S3-1b could be adjusted. The range is 10-10,000, but it doesn't appear that there are any data points below 1,000.

Please find below itemized responses to reviewers.

Reviewer #1:

Well written and methodologically complete. New method of oligo purification is presented and compares well to existing methods. Yet purpose of rather incremental increase of purity compared to other methods is not clear, especially with respect to pharmaceutical applications, where highest purity is required. The effort in developing this method seems worthwhile, further improvement can be expected. Description of experimental protocols seems complete.

We thank the reviewer for his/her support of our work. We would like to emphasize that while the oligo purity improvement is incremental, the ability to normalize concentrations while simultaneously purifying hundreds of oligos is qualitatively novel.

The acronym SNAP may interfere with identical acronyms which are eventually protected by trademarks.

We thank the reviewer for alerting us to this issue. We have renamed our method SNOP (Stoichiometrically Normalizing Oligonucleotide Purification), to avoid possible confusion with SNAP-tags.

Supplement "Methods" page 1: SNAP protocol 2nd paragraph, last word should read "removed"

Fixed, thanks.

Reviewer #2:

While this paper seems pretty novel, there is one publication that may compromise the novelty of this paper. Highly parallel oligonucleotide purification and functionalization using reversible chemistry Nucleic Acids Research, 2012, Vol. 40, No. 1, Kerri T. York, Ryan C. Smith, Rob Yang, Peter C. Melnyk, Melissa M. Wiley, Casey M. Turk, Mostafa Ronaghi, Kevin L. Gunderson, and Frank J. Steemers, from Illumina.*

We thank the reviewer for pointing out this paper, and have included it in our list of references. We do believe that the SNOP method we present exhibits several advantages over the work by York et al., namely in that SNOP normalizes the stoichiometry of oligos in the reaction. Furthermore, the York manuscript claims a 90-99% purity based only on gel data, which is significantly less accurate than the NGS approach we use here to systematically characterize oligo purities. Consequently, we believe that the SNOP products may be more pure than the York technique, though we did not explicitly verify this (our lab is not expert at the His-Tag approach, so our results may not be comparable to the original authors').

This paper described a "catch release" technology that can generate 5'-biotin functionalized oligos product with the following benefits

1. *High throughput, up to 13000-15000 oligos, in comparison, the manuscript proved up to 256 oligos experimentally, further throughput increase may not be that easy (I will elaborate separately).*

As an academic research group, our budgets for oligonucleotide synthesis are significantly lower than that of a large company. Synthesis cost for a set of 64 oligos (including HPLC- and PAGE-purified ones for purity comparison) was roughly \$13,000, and our set of 256 oligos was another \$5000 or so. We believe that the major contribution of our manuscript is the conceptual novelty of a new method that not only purifies oligo pools with better purity than HPLC and PAGE, but also normalizes oligo stoichiometries.

2. Scalable, up to μmol for oligo pools, in comparison, the manuscript proved pmol for oligo pools.

We thank the reviewer to pointing out this potential concern. To address this, we have performed new SNOF experiments on the 1.5 nmol scale. Because we do not see any significant difference in the purity and the stoichiometry of the SNOF products across this 100-fold scale variation, we feel optimistic that SNOF will function similar when applied to even larger (e.g. μmol) scales. Again, the reason we did not test even larger scales is the cost of oligo synthesis (we would have needed to reorder all precursor oligo at a larger synthesis scale).

New figure displayed in Supporting Information Section S5 on scaling SNOF.

3. Purification, by nature the catch release technology eliminates most of the oligo failures that do not have 5'-aldehyde mods, it may not eliminate internal deletion oligo failures that do have 5'-aldehyde mods, but such failures are largely dependent on raw oligo synthesis quality. In comparison, the manuscript claimed to purify off internal deletion oligo failures (to certain extent).

The reviewer interprets our claim correctly, we believe that SNOF products exhibit fewer **internal** errors as compared to the precursors, even though we do not directly interact with that sequence. The reason for this is because synthesis errors are correlated. To clarify this point for the reviewer and for readers, we have performed new NGS experiments to show the correlation between truncations in the tag and the internal sequence errors in the product sequence, shown below and also inserted into manuscript Fig. 1. Because of the correlated nature of synthesis errors, we believe that SNOF products are higher in purity than standard affinity tag-based oligo purification methods.

New figure displayed in manuscript Fig. 1 on correlation between synthesis errors.

4. *Equal concentration in the end product, in the paper and in our production, we have not evaluated experimentally if equal concentration in the end product is the case, but it does not matter to ILMN's exome enrichment application as the final 5'-biotin oligos are used in large excess to capture exome targets. In comparison, the manuscript claimed equal concentration in final product, I would like to understand more on why equal concentration is important with examples that matter to specific applications. In other words, why do we care about equal concentration?*

We would like to point out that SNOP not only can normalize product concentrations to roughly equal levels, but also allows custom adjustment of subsets of oligos to higher or lower concentrations. As far as we understand, Illumina, Nimblegen, Agilent, and Twist Biosciences are not currently pursuing concentration control in oligos used in hybrid-capture panels. We believe this likely contributes to non-uniformity in capture yield across the exome, because our studies on hybridization kinetics (Zhang et al., *Nature Chemistry* 2018, "Predicting DNA hybridization kinetics from sequence") show that probe hybridization rate constants differ by up to 3 orders of magnitude. Consequently, it is almost certain that in a standard 16 hour hybrid capture protocol with 50 pM probes, not all probe hybridizations have reached saturation. Tunable oligo stoichiometries offers a way to increase the concentrations of slow-binding probes, potentially resulting in more uniform exome coverage.

As a point of scientific philosophy, we would also like to point out that technology development follows a feedback between **novel capabilities** and **novel applications**. We view the current SNOP manuscript as an advance in capability; because the capability did not previously exist, novel applications to take advantage of this capability may not be obvious.

5. *Non-specific binding. Non-specific binding is not a concern to catch release paper due to high concentration UREA usage, in comparison, non specific binding might be a concern to the hybe-immob-wash-cut approach because (1) streptavidin coated magnetic beads are WELL known to have non-specific binding (2) undesired non-specific duplex formation could increase as throughput or plex increases*

We agree with the reviewer on this point: we have noticed significant non-specific binding not only to magnetic beads, but also to the walls of our plasticware; these are only partially mitigated by passivation strategies and reagents. Our new Fig. 1c (reproduced below) shows that a precursor with a tag truncated by 8 nt has roughly 200-fold lower capture yield than a precursor with perfect tag. We believe that this dominantly due to non-specific binding – in other words, there is a baseline binding yield of about 0.5% for all oligos regardless of sequence.

However, we disagree with the reviewer regarding the impact of the non-specific binding – even with the nonspecific

binding, SNOP is higher purity than HPLC and PAGE, and likely also than the affinity-tag method.

New figure displayed in manuscript Fig. 1 on capture yield against different tag variants.

This paper can be interesting to others in the field and can influence thinking if the authors can specifically identify applications that are currently difficult to attain but would be otherwise easy to accomplish with the hybe-immob-wash-cut approach. There are, however, several limitations in this manuscript that may limit its influence.

We hope that our response above and below convince the reviewer of the utility of our approach.

1. As the way it is right now, the 5' end cannot bear useful functionalization. If the authors switch to 5'-biotin capture probes (see below section), then 3'-end cannot bear useful functionalization, this could be a limitation. In comparison, the catch release technology can have both 3'-and 5'- functionalization separately, if required by certain applications.

The reviewer is correct in that one limitation of SNOP is that the products cannot be dual functionalized at both the 5' end and 3' end. Consequently, the SNOP products cannot be Taqman probes. However, we believe that SNOP will still be valuable for the many applications of oligo usage that require either unfunctionalized oligos, or oligos functionalized only at 1 end.

2. This approach added 20 or more bases to the required synthesis. Thus, we expect it to be functionally limited to synthetic pools of <80 bp in length, possibly <60 bp in a practical sense.

In our experience interacting with oligo suppliers such as IDT and Sigma, synthesis of oligonucleotides up to 120 nt are routine, and oligos up to 150 nt can be requested. Thus, SNOP product lengths are limited to roughly 100 nt for standard synthesis, and 130 nt if pushed for lengths.

3. Scale is limited with regard to synthesis pools. Likely this technique is useful only for smaller quantities, especially based on the necessity to oversynthesize b/c the expensive pull-down streptavidin coated magnetic beads are the limiting reagent.

We have also demonstrated SNOP with much more economical streptavidin agarose resin beads, which are roughly 20-fold less expensive than magnetic beads.

New figure displayed in Supplementary Section S5 on higher-scale SNOP using agarose resin beads.

4. *The hybe-immob-wash-cut approach needs to use excessive oligos to saturate capture probes with duplex formation on desired targets, it is definitely not inexpensive, especially when scaling up, cost becomes a concern. In addition, the cost of USER enzymes can be a concern when scaling up.*

We disagree with the reviewer on this point. HPLC and PAGE purification typically loses at least 75% of the oligos, based on our experience ordering more than 10,000 oligos from IDT over the past decade in standard desalted form vs. HPLC- or PAGE-purified form. The dominant cost in the HPLC- or PAGE-purified oligos is always labor or equipment amortization, and not in the loss of the impure oligo precursors. It is also worth mentioning that the York paper that the reviewer compares our SNOP work to also reports a yield of only 16%.

The deoxyuracil and the USER enzyme can be replaced with a photolabile chemical moiety; this is a commonly used chemistry that can be used to replace the USER cleavage if USER enzyme mix is found to be a significant component of the cost.

5. *Bioinformatic intensive for barcode design and no guarantee that cross-hybing will not be a factor for novel, new sequences.*

We have essentially solved the bioinformatic intensive barcode design process, and this is part of the novelty of our paper. We agree that there is no guarantee that cross-hybing will not be a factor for novel, new sequences. However, there is also no guarantee that novel, new sequences can be synthesized by phosphoramidite chemistry in the first place, or that novel, new sequences can be sequenced by the Illumina NGS platforms.

In general some claims seem reasonably convincing, for example, purity increase in general, although more control experiments would be needed.

We have performed more control experiments as requested by the reviewer including comparisons against HPLC, use of other functionalizations such as carboxyl EDC for allowing biotin-functionalized oligos, and analysis of precursor sequence error correlations. The new results are shown here and also in our updated manuscript (Fig. 1) and supplementary information (Sections 2 and 5).

The “scaling up is technically straightforward” seems questionable in discussion session. Another paragraph in discussion session may actually seem to explain why scaling up can be technically challenging.

We have performed larger scale SNOP purifications as requested by the reviewer (see comment 2 below). Our experiments on products at the 1.5 nmol scale is roughly a factor of 100 higher than our typical 20 pmol scale, and the results in both purity and stoichiometry are essentially indistinguishable. These results are displayed in Supplementary Fig. S5-1.

I would like to get into technical details in this section: 1. 3'-biotin capture probe(s) preparation was specified as "made from IDT." I am going to assume it's made from 3'-biotin CPG in oligo synthesis. 3'-Biotin oligos made from 3'-biotin CPG are known to have stability issues, the 3'-biotin may come off really quick due to hydrolysis in solution (I can explain in details if needed). That said, I wonder if the authors have experienced such instability issue. In general, efforts to reproduce the hybe-immob-wash-cut approach can be hard with a known stability issue. I suggest the authors try two things: (1) make 3'-biotin oligos from a different way (there are different ways) (2) try 5'-biotin capture oligos

We have experimentally tested both of the reviewer's suggestions, and the results are shown here and displayed in Supplementary Fig. S5-3. As the reviewer can see, there does not appear to be a significant different in the performance for different biotin functionalizations.

Effects of different Biotin functionalizations on SNOP.

2. The hybe-immob-wash-cut approach needs to use excessive oligos to saturate capture probes with duplex formation on desired targets, so N-1 impurity or N-2 impurity do not form duplex, but there are no experiments to prove that's the case. I would expect at least one experiment with N-1 spike in in the 1-plex experiment with LCMS analysis to show N-1 spike does not exist in final product.

We have done the experiments requested by the reviewer, and these are in the new Fig. 1c. The tag with the 1nt truncation is captured with roughly 3-fold lower yield than the perfect tag, and the tag with the 2nt truncation is captured with 38-fold lower yield than the perfect tag.

3. For purity increase, it would be nice to add HPLC purified oligos (either reverse phase HPLC purified oligos and/or anion-exchange HPLC oligos) as one more control, in addition to PAGE purified oligos.

We have performed new experiments on HPLC-purified oligos, shown below and inserted in Supplementary Info Section S2.

New figure displayed in supplementary section S2 on 64-plex SNOP vs. individually HPLC-purified oligos.

Please see my previous comments about one publication missing from the manuscript.

We have included a reference to the publication mentioned by the reviewer.

I am not sure if a resubmission in the future in the future should be considered for *Nature Communications*, mainly due to novelty compromised. If the authors could identify some particular applications that are currently different to attain but would be otherwise easy to accomplish with the hybe-immob-wash-cut approach. Such applications need to be fairly useful to most users in genomic field and/or biotech field.

We disagree with the reviewer about the novelty of our paper. We have not seen convincing evidence that the previous affinity-tag method is higher purity than HPLC or PAGE, whereas our SNOP products, under rigorous NGS analysis, is significantly higher purity. Additionally, the stoichiometric control is a qualitatively novel aspect of the technology never before proposed or demonstrated.

Reviewer #3:

In the manuscript "Simultaneous and Stoichiometric Purification of Hundreds of Oligonucleotides," the authors introduce a novel method for the purification and balancing of pools of crude oligonucleotides. Purification beads are coated with a capture oligo containing a constant (?universal?) dodecamer, a degenerate (?barcode?) octamer, and a single RNA base (deoxyuracil). Crude oligo pools are synthesized that include the universal sequence at the 5' end. The pools are hybridized to the beads and then cleaved enzymatically. The carefully balanced degenerate octamer serves to balance the concentrations of the individual oligos in the pool. Testing of the method produced oligo purities around 80% with elements balanced in a two-fold range around the median. This is very good performance.

We thank the reviewer for his/her support of our work.

The discussion section compares the SNAP method only with PAGE-purification methods. It would be useful to also compare and contrast SNAP with enzymatic mismatch cleavage/removal methods. For example, the use of *RecJ* has been evaluated for use on conventional single-stranded oligos (Pubmed: 28459543), while *MutS* and other enzymes have been used for error-correction on double-stranded products built from conventional and microarray oligos (Pubmed examples: 15561997, 28733633).

We thank the reviewer for the suggestions, have added the suggested articles to our references list. Additionally, we have performed new experiments comparing SNOP (new acronym at Reviewer 1's suggestion) with HPLC-purified oligos; these appear in Supplementary Fig. S2-3. However, we do not have much experience with enzymatic mismatch cleavage/removal methods suggested; consequently, we do not feel that we can competently apply the techniques to multiplexed oligo purification within the timeframe of this revision, considering that significant method development may be needed to adapt some of the techniques to single-stranded oligos.

The figures are all clear, appropriate, and support the conclusions of the paper. Perhaps the scale on supplemental figure S3-1b could be adjusted. The range is 10-10,000, but it doesn't appear that there are any data points below 1,000.

We thank the reviewer for the suggestion and have made the figure change.

REVIEWERS' COMMENTS:

Reviewer #1 (Remarks to the Author):

Questions were answered adequately and proposed changes were introduced.

Reviewer #2 (Remarks to the Author):

In the revised manuscript, the authors have done a good job in adding control experiments that were missing. Kudos on that.

Like I mentioned before, the novelty of this manuscript seems to be compromised by a 2012 publication in *Nucleic Acids Research*, 2012 (Highly parallel oligonucleotide purification and functionalization using reversible chemistry *Nucleic Acids Research*, 2012, Vol. 40, No. 1, Kerri T. York, Ryan C. Smith, Rob Yang, Peter C. Melnyk, Melissa M. Wiley, Casey M. Turk, Mostafa Ronaghi, Kevin L. Gunderson, and Frank J. Steemers).

For the sake of comparison, I would like to compare the novelty side by side to make it clear.

Novelty claimed in the revised manuscript Novelty published in *Nucleic Acids Research*, 2012 by York et al.

1. Simultaneously purification of hundreds of oligos

Simultaneously purification of ~13000 oligos

2. With improved purity over HPLC/PAGE, demonstrated by NGS analysis

the improved purity of the product was demonstrated by FPLC (as opposed to NGS analysis), but the product have been used for NGS exome enrichment library prep process for NGS application

3. Concentration normalization of final product—"a capability never before achieved in multiplex purification" as claimed by the authors

The end of this publication pointed out that "Such an approach would allow concentration normalization across individual oligonucleotides" (claimed at the end of the publication)

In summary, the revised manuscript describes a nice alternative approach to simultaneously purify oligos in low plexity (hundreds of oligos) with improved purity and concentration normalization feature. However, the novelty claimed in this manuscript may not appear to be novel enough for *Nature Communication*.

Reviewer #3 (Remarks to the Author):

In the manuscript "Simultaneous and Stoichiometric Purification of Hundreds of Oligonucleotides," the authors introduce a novel method (SNOP) for the purification and balancing of pools of crude oligonucleotides. Purification beads are coated with a capture oligo containing a constant (universal) dodecamer, a degenerate (barcode) octamer, and a single RNA base (deoxyuracil). Crude oligo pools are synthesized that include the universal sequence at the 5' end. The pools are hybridized to the beads and then cleaved enzymatically. The carefully balanced degenerate octamer serves to balance the concentrations of the individual oligos in the pool. Testing of the method produced oligo purities around 80% with elements balanced in a two-fold range around the median. This is very good performance.

Any concerns from the earlier draft have been addressed in this manuscript. The introduction compares the SNOP method with PAGE and enzymatic purification methods and there is sufficient experimental comparison. The figures are all clear, appropriate, and support the conclusions of the paper.